# GCP-VQVAE: A GEOMETRY-COMPLETE LANGUAGE FOR PROTEIN 3D STRUCTURE

## ABSTRACT

Converting protein tertiary structure into discrete tokens via vector-quantized variational autoencoders (VQ-VAEs) creates a language of 3D geometry and provides a natural interface between sequence and structure models. While pose invariance is commonly enforced, retaining chirality and directional cues without sacrificing reconstruction accuracy remains challenging. In this paper, we introduce GCP-VQVAE, a geometry-complete tokenizer built around a strictly SE(3)-equivariant GCPNet encoder that preserves orientation and chirality of protein backbones. We vector-quantize pose-invariant readouts into a 4 096-token vocabulary, and a transformer decoder maps tokens back to backbone coordinates via a 6D rotation head trained with SE(3)-invariant objectives.

Building on these properties, we train GCP-VQVAE on a corpus of 24 million monomer protein backbone structures gathered from the AlphaFold Protein Structure Database. On the CAMEO2024, CASP15, and CASP16 evaluation datasets, the model achieves backbone RMSDs of 0.4377 Å, 0.5293 Å, and 0.7567 Å, respectively, and achieves 100% codebook utilization on a held-out validation set, substantially outperforming prior VQ-VAE–based tokenizers and achieving state-of-the-art performance. Beyond these benchmarks, on a zero-shot set of 2 261 completely new experimental structures, GCP-VQVAE attains a backbone RMSD of 0.8033 Å and a TM-score of 0.9747, demonstrating robust generalization to unseen proteins. Lastly, we elaborate on the various applications of this foundation-like model, such as protein structure compression and the integration of generative protein language models. We make the GCP-VQVAE source code, zero-shot dataset, and its pretrained weights fully open for the research community.

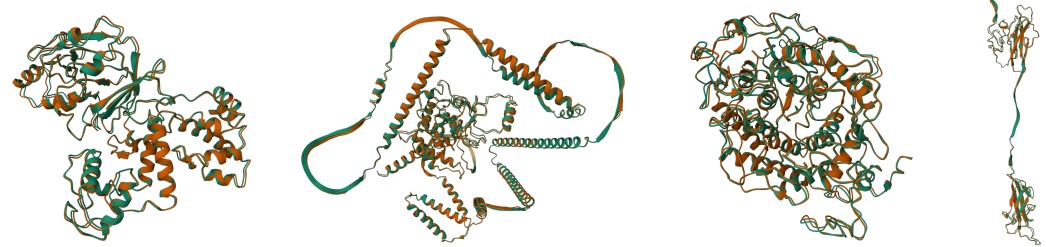

Figure 1: Superposition of GCP-VQVAE reconstructions (orange) with native structures (green) for four long, held-out backbone coordinates of proteins. Left→right: T1079 (CASP14; 482 residues; TM-score 0.9909; RMSD 0.7533 Å), T2272S8 (CASP16; 818 residues; TM-score 0.9967; RMSD 0.5639 Å), T1157s1-D1 (CASP15; 661 residues; TM-score 0.9920; RMSD 0.8172 Å), and 8BQU, chain A (CAMEO2024; 298 residues; TM-score 0.9764; RMSD 0.9981 Å).

## 1 INTRODUCTION

Proteins are the molecular machines of life, and their function is intricately tied to their three-dimensional structures (Bertoline et al., 2023; Tripathi et al., 2025). Understanding and predicting

these structures remains one of the central challenges in computational biology (Jänes & Beltrao, 2024). Just as natural language is governed by grammatical and contextual rules, protein 3D structures exhibit spatial patterns and constraints that suggest an underlying "grammar" of folds and interactions (Ruff et al., 2022; Kilgore et al., 2025; Weissenow & Rost, 2025).

Despite advances in protein structure prediction, effectively representing the 3D geometry of proteins in a form suitable for generative modeling remains an open problem (Draizen et al., 2024; Lu et al., 2025). While recent methods have begun to leverage generative AI, such as diffusion models and autoregressive frameworks, to produce full-atom structures or backbone coordinates, they still lag behind other domains such as language or vision in terms of reconstruction precision, scalability to large and diverse datasets, and openness for the broader research community, like natural language or image and video generation (Yuan et al., 2025b). These gaps continue to constrain our ability to build powerful, general-purpose protein models using modern AI techniques.

Beyond generative modeling, learning a discrete language for protein 3D structure opens up a wide range of downstream applications. First, compressing 3D coordinates into compact sequences of integer codes—while preserving accurate reconstruction—can substantially reduce storage and transmission costs for structural data (Kim et al., 2023). Second, discrete structural representations enable fast, alignment-style comparison of protein shapes, analogous to multiple sequence alignment in sequence space (Van Kempen et al., 2024). Third, in learnable quantization frameworks such as vector-quantized autoencoders, these codes can be decoded into semantically rich continuous embeddings (Yuan et al., 2025b), facilitating structure-aware feature extraction for classification, clustering, structure-based comparison/search, and other predictive tasks. Finally, unifying protein sequence and structure through a shared discrete representation may pave the way for multimodal generative models that bridge amino acid sequences and 3D folds within a common language modeling framework (Hsieh et al., 2025).

However, most existing protein-structure VQVAEs are either closed-source or only partially released (e.g., code without full evaluation scripts or strongest checkpoints), which impedes reproducible comparison (Gao et al., 2024c; Hayes et al., 2025). Furthermore, the publicly available baselines often generalize weakly to *unseen* proteins. Consequently, the field lacks a fully open-source, high-accuracy tokenizer with transparent training and evaluation that demonstrably transfers to new proteins.

**Contributions.** (1) We introduce GCP-VQVAE, a geometry-complete tokenizer that preserves orientation and chirality while producing pose-invariant codes that support missing coordinates. (2) We scale training to 24M monomers and report exhaustive evaluations (CAMEO2024, CASP14/15/16) and a zero-shot suite of newly deposited experimental structures. (3) Our model achieves state-of-the-art reconstruction on a diverse range of unseen 3D structures, and stays ahead of other open-source methods. On the zero-shot set, GCP-VQVAE attains a backbone RMSD of 0.8033 Å and a TM-score of 0.9747. (4) We release code, checkpoints, and unified evaluation scripts for ourselves and baselines, enabling reproducible comparison.

## 2 RELATED WORK

An early and influential approach to casting protein 3D structure as a discrete language is FoldSeek van Kempen et al. (2022), which learns a 20-state 3Di alphabet with a VQ-VAE trained for evolutionary conservation and encodes structures as token sequences for ultra-fast k-mer–based local/global alignment. Building on the notion of a discrete structural language—but targeting generative reconstruction rather than search—the FoldToken series (Gao et al., 2025; 2024a;b;c) develops a VQ-VAE–style tokenizer and decoder: FoldToken introduces a SoftCVQ fold language with joint sequence–structure generation; FoldToken2 stabilizes quantization and extends to multi-chain settings; FoldToken3 mitigates gradient/class-space issues to reach 256-token compression with minimal loss; and FoldToken4 unifies cross-scale consistency and hierarchies in a single model, reducing redundant multi-scale training and code storage.

Yuan et al. (2025b) introduces AminoAseed codebook reparameterization plus Pareto-optimal K×D sizing, proposes structtokenbench—a fine-grained evaluation suite—and diagnoses codebook under-utilization in VQVAE PSTs. Concurrently, Gaujac et al. (2024) tokenizes protein backbones with

a VQ autoencoder (codebooks 4k–64k); its main open-source limitation is that it does not support sequences shorter than 50 or longer than 512 amino acids.

ESM-3 (Hayes et al., 2025) couples its multimodal transformer with a VQVAE structure tokenizer that discretizes local 3D geometry into structure tokens; structure, sequence, and function are jointly trained under a masked-token objective. The tokenizer uses an SE(3)-aware module within the encoder, and ESM-3 employs a 4 096-code structure codebook (plus special tokens) for downstream generation and masked reconstruction.

Across prior structure tokenizers, neither openness nor accuracy are yet satisfactory. The FoldTo-ken line spans four variants, but to our knowledge, only FoldToken-4 offers a partial open release, without the strongest checkpoints, limiting reproducibility (Gao et al., 2024c). ESM-3 exposes an internal VQ-VAE tokenizer, yet public artifacts lack fully documented training/evaluation proce-dures and best weights, making directly comparable reconstruction benchmarking difficult (Hayes et al., 2025). The open VQ autoencoder of Gaujac et al. (2024) supports a restricted length window (e.g., ∼50–512 residues), precluding fair assessment on long chains where reconstruction accuracy degradation is most evident. Finally, FoldSeek's learned 3Di alphabet targets ultra-fast structure search and does not provide a generative decoder from discrete codes back to coordinates, so re-construction fidelity cannot be evaluated (van Kempen et al., 2022). Consequently, the community still lacks a fully open, end-to-end tokenizer with released best weights and source codes that attains high reconstruction accuracy on *unseen* proteins.

## 3 DATASET

We began with the latest release of UniRef50 (Consortium, 2019) [1], which clusters protein sequences at 50% identity and thus offers a natural, non-redundant scaffold for large-scale structure modeling. For every UniRef50 entry with a corresponding model in the AlphaFold Database (AFDB; Varadi et al. (2022)), we downloaded the per-protein structure (PDB format at collection time). Limiting to UniRef50 reduces near-duplicate leakage by ensuring that homologs within clusters do not exceed 50% identity. After parsing and splitting multi-chain records into individual chains, this procedure yielded approximately 42M single-chain PDB samples.

From this AFDB ∩ UniRef50 pool, we drew a uniform random sample of 24M single-chain struc-tures to form the training set. This down-sampling keeps training throughput tractable while pre-serving the global distribution of lengths, folds, and taxa present in the full pool.

Table 1: Dataset and benchmark statistics. Training data is deduplicated at 100% sequence identity against all validation/test splits and external benchmarks (CAMEO2024, CASP14–16). We also include a zero-shot set of 2 261 newly deposited experimental monomer chains (PDB, 2024–2025).

| Split | Source | Selection criterion | # Samples |
|---|---|---|---|
| Training | AFDB ∩ UniRef50 | Uniform random sample from ∼42M pool | 24 000 000 |
| Validation | From Tests A/B/C | 2k random per test (A,B,C) merged | 6 000 |
| Test set A | PDB DB | cross-referenced AF pLDDT ≤ 70; len ≥ 25; < 25 consecutive missing residues; dedup | 31 112 |
| Test set B | AFDB | 97 species; ∼2.5k/species; len ≥ 25; dedup | 17 353 |
| Test set C | PDB DB | under-represented taxonomies in SwissProt; pLDDT ≥ 95; < 25 consecutive missing residues; dedup | 31 867 |
| Zero-shot (experimental) | PDB DB | recent experimental monomers (2024–2025); len 25–2048; dedup; < 50% seq id vs train | 2 261 |
| CAMEO2024 | CAMEO | - | 574 |
| CASP14 | CASP | - | 33 |
| CASP15 | CASP | - | 45 |
| CASP16 | AF3 predictions | - | 104 |

We evaluate on three held-out test suites capturing complementary shifts (see Table 1): *A* low-confidence AF2 SwissProt models, *B* species-diverse (97-species) taxonomic shift, and *C* high-confidence structures from under-represented taxa. All suites are per-chain, length-filtered, dedupli-cated, and NaN-aware; a balanced validation set of 6,000 chains (2k per suite) is sampled disjointly. Full curation criteria, thresholds, and counts are provided in Appendix A.2 .

To probe generalization to genuinely new structures, we assembled a zero-shot suite from *recently deposited experimental* monomeric chains in the PDB (calendar years 2024–2025). Multi-chain entries were split into per-chain backbones, chains shorter than 25 residues or longer than 2 048 residues were discarded, and we retained entries even when they contained missing coordinates

---

[1]March 2024 release.

(NaN) to reflect typical experimental gaps, yielding 76 503 pdbs in total. We deduplicated using structural clustering with FoldSeek `easy-cluster` (alignment type 2, tmscore threshold 0.95), retaining one representative per cluster. This resulted in 19 080 clusters (van Kempen et al., 2022). For leakage control, each representative was compared with foldseek `easy-search` against (i) the AFDB ∩ UniRef50 set used for our GCP-VQVAE training and (ii) the `afdb_rep_v4` (Barrio-Hernandez et al., 2023) representative set used to pre-train the GCPNet encoder in ProteinWorkshop (Jamasb et al., 2024). We *excluded* any candidate with $\geq 50\%$ structure similarity to any chain in those sources; the resulting zero-shot set comprises 2 261 monomer chains.

**Independent benchmarks.** In addition to the internal splits, we evaluate on community benchmarks to facilitate comparison with prior work: CAMEO2024 (Robin et al., 2021; Leemann et al., 2023), CASP14 (Kryshtafovych et al., 2021), CASP15 (Kryshtafovych et al., 2023), and AlphaFold3 (Abramson et al., 2024) predictions for CASP16 (Yuan et al., 2025a) targets. Each benchmark is processed with the same per-chain extraction and deduplication pipeline and is used strictly for out-of-distribution evaluation.

Table 1 summarizes the composition and selection criteria of all splits. Counts after converting multi-chain inputs into per-chain samples. Also, all samples are truncated to a maximum of 2048 amino acids in length.

## 4 GCP-VQVAE ARCHITECTURE

The proposed architecture leverages two main parts: (1) a GCPNet encoder to encode backbone coordinates into embeddings, and (2) a transformer-based VQVAE, which discretizes backbone embeddings and then converts them back into 3D coordinates.

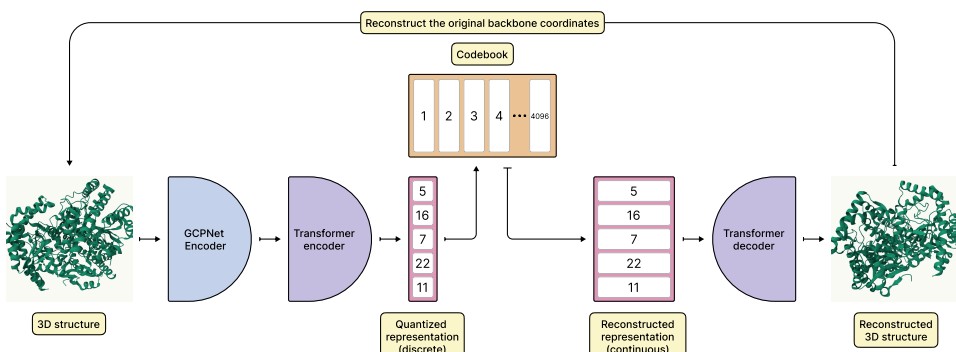

Figure 2: GCP-VQVAE overview. A protein backbone (N–$C_\alpha$–C) is first encoded by a SE(3)-equivariant GCPNet that preserves orientation and chirality. A Transformer encoder produces latents that are vector-quantized into a sequence of code indices from a 4 096-entry codebook (e.g., 5, 16, 7, 22, 11), yielding pose-invariant discrete tokens. For reconstruction, the indices are de-quantized to continuous embeddings and passed to a Transformer decoder equipped with a 6-D rotation head, which predicts rigid updates to recover the original backbone coordinates.

### 4.1 GCPNET ENCODER

GCPNet (Morehead & Cheng, 2024b) extends the scalar–vector message-passing philosophy of GVP-GNN to a geometry-complete, SE(3)-equivariant encoder. Every atom $i$ in a molecular graph $G = (V, E)$ carries scalars $s_i \in \mathbb{R}^{d_s}$ and row-wise vectors $\mathbf{v}_i \in \mathbb{R}^{d_v \times 3}$ that rotate as $\mathbf{v}_i \mapsto R\mathbf{v}_i$ under $g = (R, \mathbf{t}) \in \text{SE}(3)$, while each edge $(i, j)$ stores analogous features $(s_{ij}, \mathbf{v}_{ij})$ and the relative displacement $\mathbf{r}_{ij} = \mathbf{x}_j - \mathbf{x}_i$. Before any update, the model attaches to every edge a right-handed orthonormal frame $F_{ij} = [\mathbf{a}_{ij}, \mathbf{b}_{ij}, \mathbf{c}_{ij}] \in \text{SO}(3)$ with $\mathbf{a}_{ij} = \mathbf{r}_{ij}/\|\mathbf{r}_{ij}\|$; this frame supplies a reference for chirality and orientation that GVP-GNN lacks.

The core computation is a Geometry-Complete Perceptron (GCP) micro-step that first down-scales the vectors and then projects them into the local frame to extract nine orientation-aware features.

Denoting $\mathbf{z}_{ij} = W_d \mathbf{v}_{ij}$ and $\text{vec}(\cdot)$ the row-wise vectorization, the joint update mixes the old scalars with their orientation signatures (the frame-projected vectors and their norms) and gates the vectors through a learnable row-wise gate to preserve equivariance; $\phi_s$ is an MLP and $\sigma$ denotes a learnable gating function that need not be a fixed sigmoid (Equation 1).

$$\left(s_{ij}, \mathbf{v}_{ij}\right) \;\mapsto\; \left(\phi_s\big[s_{ij}, \|\mathbf{z}_{ij}\|_2, \text{vec}(\mathbf{z}_{ij} F_{ij}^\top)\big], \; \sigma\big(W_g s_{ij}\big) \odot W_v \mathbf{z}_{ij}\right) \tag{1}$$

In the node update, the orientation features are averaged over neighbors. A sequence of such micro-steps, wrapped by a residual shortcut, forms a GCPConv edge block. After aggregating messages $m_i = \sum_{j \in \mathcal{N}(i)} \text{GCPConv}(i,j)$, a gated scalar–vector MLP updates the node features, and stacking $L$ layers yields an invariant backbone. When tasks require coordinates, each layer appends an equivariant displacement head whose output is a learned 3-dimensional vector; this vector is added residually to $\mathbf{x}_i$ and re-centered to ensure translation invariance, enabling force or trajectory prediction without breaking equivariance.

$$\mathbf{x}_i \;\leftarrow\; \mathbf{x}_i + \mathbf{f}_i, \qquad \mathbf{f}_i = \text{MLP}_{\text{disp}}(m_i) \tag{2}$$

Equations 1 and 2 commute with every rigid motion, making the encoder strictly SE(3)-equivariant, while projection through $F_{ij}$ preserves the complete set of edge orientations so that the latent remains *geometry-complete* at any depth. Eliminating the frame ($F_{ij} = I$), replacing the orientation features by $\|\mathbf{v}_{ij}\|_2$, and dropping the coordinate head restores a frame-free, E(3)-equivariant network akin to GVP-GNN—thereby isolating the contributions responsible for our empirical gains. Because GCPNet keeps directional and chiral cues that GVP-GNN discards, it supports optional equivariant coordinate updates for force or dynamics prediction and attains improved performance across invariant (e.g., binding affinity; Morehead & Cheng (2024b)), equivariant (e.g., force regression; Morehead & Cheng (2024b)), and coordinate-generative (e.g., molecular diffusion; Morehead & Cheng (2024a)) tasks with only modest extra computation.

## 4.2 VQVAE

The transformer–based VQVAE employed in this work is organized into the classical three-stage pipeline of *encoder*, *vector-quantization*, and *decoder*. The encoder processes the given embeddings into a sequence of latent vectors, the quantizer discretizes these latents, and the decoder reconstructs the original signal from the resulting code indices.

Both stacks adopt a lightweight pre-layer-normalized Transformer that integrates several recent efficiency upgrades: (i) Pre-LayerNorm places the LayerNorm before each sub-block (Xiong et al., 2020), which keeps activations in a well-behaved range throughout the network, reduces gradient-scale drift, and therefore allows training with larger learning rates and much milder warm-up schedules; (ii) separately normalizes query and key vectors before computing attention logits (Henry et al., 2020). This prevents overly large dot-products, stabilizes attention distributions, and mitigates softmax saturation, especially in low-resource or small-batch training scenarios; (iii) Grouped-Query Attention shares key–value projections across groups of query heads (Ainslie et al., 2023), reducing both memory and compute without harming quality; (iv) Rotary Positional Embeddings (RoPE) inject relative-position information by applying position-dependent planar rotations to each query–key pair (Su et al., 2024), letting the model generalize to much longer sequences with virtually no extra computational cost; (v) We remove bias terms from projection and feed-forward layers, an established simplification that has negligible effect on accuracy while trimming a fraction of parameters and FLOPs.

In the architecture, the vector quantization layer provides the discrete bottleneck between the aforementioned encoder and decoder and follows the learnable formulation of VQVAE, augmented with several improvements described in the following to accelerate convergence, increase codebook usage, and stabilize large codebooks training.

Before training starts, we run $k$-means on the encoder outputs of the first mini-batch to seed the codebook as displayed in Equation 3, where $Z^{(0)} \subset \mathbb{R}^d$ are the features, $K$ the codebook size, and $T$ the number of Lloyd iterations. Empirically, this step mitigates early code collapse and improves utilization when $K$ is large.

$$E^{(0)} = \text{KMEANS}(Z^{(0)}, K, T), \tag{3}$$

Given an encoder vector $\mathbf{z} \in \mathbb{R}^d$, quantization proceeds by nearest-neighbor lookup, Equation 4, which is identical to the vanilla VQ rule.

$$k = \arg\min_{j} \|\mathbf{z} - \mathbf{e}_j\|_2^2, \qquad \mathbf{z}_q = \mathbf{e}_k, \tag{4}$$

To transmit gradients through this non-differentiable operation, instead of using the straight-through estimate (STE) introduced in Van Den Oord et al. (2017), we adopt the rotation trick (Fifty et al., 2024). During back-propagation, the Jacobian is replaced by Equation 5, where $R$ is the shortest-arc rotation aligning the unit vectors $\hat{\mathbf{z}}$ and $\hat{\mathbf{e}}_k$. This modification embeds both angular and magnitude mismatch into the back-propagated signal, yielding faster convergence and richer code usage in practice.

$$\mathbf{J}_k = \frac{\|\mathbf{e}_k\|}{\|\mathbf{z}\|} R(\hat{\mathbf{z}} \to \hat{\mathbf{e}}_k), \tag{5}$$

The codebook updates and the encoder commitment follow the standard VQ losses (Equation 6), with $\beta$ balancing the commitment pressure.

$$\mathcal{L}_{\text{code}} = \|\text{sg}[\mathbf{z}] - \mathbf{e}_k\|_2^2, \qquad \mathcal{L}_{\text{commit}} = \beta \|\mathbf{z} - \text{sg}[\mathbf{e}_k]\|_2^2, \tag{6}$$

To encourage diverse and well-separated embeddings, we regularize the codebook with the Frobenius norm (Equation 7) weighted by $\lambda_{\text{orth}}$. This orthogonality constraint spreads codes over the hypersphere and curbs under-utilization, which is a common failure mode in large books.

$$\mathcal{L}_{\text{orth}} = \|E^\top E - I_K\|_F^2, \tag{7}$$

To translate the decoder's abstract embeddings into a physically meaningful 3D structure, we utilize a 6D rotation head on top of the decoder. This module's primary purpose is to provide a stable and continuous parameterization of 3D rotations and translations, which is crucial for effective training of deep neural networks. This approach, notably used in AlphaFold2 and supported by the findings of Zhou et al. (2019) on rotation representations, avoids the well-known issues of Gimbal lock in Euler angles and the double-cover ambiguity of quaternions.

Intuitively, the head operates by predicting an intermediate representation for each residue, comprising two 3D direction vectors and a translation. The direction vectors are deterministically converted into a stable rotation matrix via the Gram-Schmidt process, while the translation is scaled by a hyper-parameter, $\alpha$, to an arbitrary range (e.g., Å). This resulting rigid transformation acts as an update, which is composed with the residue's running pose. The final backbone coordinates are then generated by applying this new, refined pose to a fixed local atomic template ($\mathbf{X}^{\text{local}}$). This iterative process ensures the structure is built in a geometrically consistent and equivariant manner, with the full operational details provided in Algorithm 1.

The decoder is supervised with a weighted sum of three geometric objectives. Building on the loss terms defined in Algorithm 2 (distance, direction and aligned MSE) and the Kabsch alignment returned by Algorithm 3, we supervise the decoder with the weighted sum of Equation 8 as the reconstruction part ($L_{\text{rec}}$) of the final loss.

$$\mathcal{L}_{\text{rec}} = \lambda_{\text{mse}} \mathcal{L}_{\text{MSE}} + \lambda_{\text{dist}} \mathcal{L}_{\text{dist}} + \lambda_{\text{dir}} \mathcal{L}_{\text{dir}}, \tag{8}$$

Here, $\mathcal{L}_{\text{MSE}}$ is the mean-squared error between predicted and native backbones after Kabsch alignment, $\mathcal{L}_{\text{dist}}$ penalises deviations in the $3L \times 3L$ backbone distance matrix (clamped at 5 Å$^2$), and $\mathcal{L}_{\text{dir}}$ measures squared differences of pairwise dot-product tensors over the six orientation vectors per residue (clamped at 20). Finally, the overall objective optimized for each iteration is demonstrated in Equation 9 where $\mathcal{L}_{\text{rec}}$ is the decoder reconstruction.

$$\mathcal{L} = \mathcal{L}_{\text{rec}} + \mathcal{L}_{\text{code}} + \lambda_{\text{commit}} \mathcal{L}_{\text{commit}} + \lambda_{\text{orth}} \mathcal{L}_{\text{orth}}, \tag{9}$$

# 5 EXPERIMENTS

In this section, optimization is used with AdamW (Loshchilov, 2017) and a cosine-annealed learning-rate schedule with warm-up steps (Loshchilov & Hutter, 2016). Experiments were implemented in PyTorch 2.7 (Ansel et al., 2024) [2] with mixed-precision (BF16) training (Kalamkar et al., 2019; Micikevicius et al., 2017) on four nodes of NVIDIA $8 \times$ A100 GPUs. For the GCP-Net encoder, we initialized from the ProteinWorkshop checkpoint (Jamasb et al., 2024). The full GCP-VQVAE configuration and exact hyperparameter values are listed in Appendix A.3; Table 5 and 6.

Table 2: Comparison with the available open-source structure tokenizer methods. Except for FoldToken-4, which uses 256 vocab size, all methods use 4 096 vocab size. The Structure Tokenizer of Gaujac et al. (2024) only supports sequences of length 50–512, metrics for that baseline exclude out-of-range samples (other methods are evaluated on the full sets). Also, zero-shot results for it are omitted due to limited coverage.

| Dataset | Metric | GCP-VQVAE (Ours) | FoldToken 4 (Gao et al., 2024c) | ESM-3 VQVAE (Hayes et al., 2025) | (Gaujac et al., 2024) |
|---|---|---|---|---|---|
| CASP14 | TM-score ↑ | **0.9890** | 0.5410 | 0.5042 | 0.3624 |
| | RMSD ↓ | **0.5431** | 8.9838 | 10.4611 | 10.5344 |
| CASP15 | TM-score ↑ | **0.9884** | 0.3289 | 0.3206 | 0.2329 |
| | RMSD ↓ | **0.5293** | 14.6702 | 13.1877 | 14.8956 |
| CASP16 | TM-score ↑ | **0.9857** | 0.8055 | 0.7685 | 0.6058 |
| | RMSD ↓ | **0.7567** | 5.5094 | 8.2640 | 8.7106 |
| CAMEO2024 | TM-score ↑ | **0.9918** | 0.4784 | 0.4633 | 0.3575 |
| | RMSD ↓ | **0.4377** | 12.1089 | 12.1138 | 13.5360 |
| Zero-Shot | TM-score ↑ | **0.9747** | 0.3324 | 0.3131 | - |
| | RMSD ↓ | **0.8033** | 17.4449 | 18.9335 | - |

Training proceeds in two stages on the same training split. In Stage 1, sequences are truncated to 512 residues. In Stage 2, the maximum length is increased to 1 280 residues. In Stage 2, we also introduce a simple NaN-masking augmentation to handle missing coordinates; implementation details are provided in Appendix A.3 (Figure 5). At the end of Stage 2 training, on the validation set, the model attains MAE 0.2239, RMSD 0.4281, GDT-TS 0.9856, and TM-score 0.9889 with 100% codebook utilization; per-test-set results appear in Table 3.

Table 3: Evaluate our GCP-VQVAE method on the predefined test sets.

| Method | Test set A | Test set B | Test set C |
|---|---|---|---|
| RMSD ↓ | 0.5124 | 0.4027 | 0.4787 |
| TM-score ↑ | 0.9823 | 0.9951 | 0.9885 |

We compare against open-source structure tokenizers: FoldToken-4, ESM-3 VQ-VAE, and the Structure Tokenizer of Gaujac et al. (2024). In practice, ESM-3 does not provide documented usage or official evaluation scripts for its VQ-VAE; we therefore reproduced its evaluation using the publicly released weights and their GitHub codebase. The original FoldToken-4 evaluation repository was prohibitively slow, so we re-implemented it, yielding a $\sim 20 \times$ speed-up with negligible loss in reconstruction rate. For Gaujac et al. (2024) we used the authors' released scripts with the 4 096-entry codebook checkpoint. See Table 2 for aggregate metrics; Figure 4 visualizes the full RMSD error across test sets.

We examined how reconstruction error varies with sequence length on the validation set (Appendix A.3; Figure 6). Errors increase only mildly with length, indicating stable accuracy up to 1 280 residues with only a slight variance rise for very long chains. On the zero-shot set we obtain mean RMSD 0.8033 Å and TM-score 0.9747 (Table 2), with group-wise trends summarized in Table 4; extended zero-shot analyses appear in Appendix A.4.

---

[2] We built the VQVAE components extensively by using the x-transformers and vector-quantize-pytorch libraries.

Table 4: Metrics by structure group on the zero-shot dataset. Groups are defined by secondary-structure composition thresholds and are not mutually exclusive. Thresholds of each groups are described in Appendix A.2

| Group | Count | RMSD ↓ | TM-score ↑ |
|---|---|---|---|
| All rows | 2261 | 0.8033 | 0.9747 |
| A (mainly $\alpha$) | 615 | 0.7671 | 0.9751 |
| B (mainly $\beta$) | 582 | 0.6713 | 0.9737 |
| AB (mixed) | 97 | 0.6283 | 0.9883 |
| D (coil-rich/disordered) | 691 | 0.9004 | 0.9681 |
| No-NaN residues | 1314 | 0.5978 | 0.9796 |
| Has-NaN residues ($\geq$1) | 947 | 1.0885 | 0.9680 |

## 6 DISCUSSION AND FUTURE WORK

GCP-VQVAE delivers high-fidelity reconstruction across diverse suites of benchmarks, with TM-scores typically $\geq$ 0.98 and mean RMSDs in the 0.40–0.80 Å range on CAMEO2024, CASP14/15/16, and comparable performance on our internal Test A/B/C splits (e.g., 0.40–0.51 Å RMSD).

On our diverse validation set, although overall reconstruction is strong, we observe a mild RMSD drift with sequence length (Figure 6). We attribute this to length imbalance in training; short chains dominate, leaving the model relatively underexposed to long-range constraints and rare structural motifs. Mitigations include targeted fine-tuning on a length-balanced subset, reweighting/upsampling long sequences, and length-aware objectives, which should reduce the slope without architectural changes.

On the zero-shot set of 2 261 newly deposited experimental monomers, the model maintains strong generalization with mean RMSD 0.8033 Å and TM-score 0.9747 (Table 4). Our diagnostics indicate that degradation is driven primarily by the ratio and count of missing coordinates (Appendix Figure 10, 9), and secondarily by sequence length (Appendix Figure 8), Similar to our interpretation of the zero-shot results. Consistent with this, sequences without missing residues exhibit substantially lower errors than those with gaps (Table 4). These trends suggest further gains from strengthening the NaN-masking augmentation (e.g., longer and multi-gap patterns) and increasing exposure to longer chains.

Relative to available open-source tokenizers, our method shows a large margin (often an order of magnitude lower RMSD) while maintaining near-ceiling TM-scores. We believe part of this disparity stems from incomplete public releases: several baselines either restrict sequence lengths or withhold their strongest checkpoints (as is evident for FoldToken-4 repository) and end-to-end evaluation scripts. To enable rigorous verification, we release our checkpoints and evaluation pipelines for both GCP-VQVAE and reproduced baselines, and we invite the community to run, audit, and extend these comparisons.

For the proposed GCP-VQVAE architecture, we see the following applications:

**Structure compression.** Using a 4 096-entry codebook yields about $\sim 24\times$ compression of backbone coordinates for a 512-residue monomer (Appendix A.5). Given our low RMSDs of $\sim 0.4$–$0.8$ Å (Table 2), this is a practical lossy backbone codec. Further gains are possible by (i) allowing a single code to represent short residue spans (e.g., 2–4 amino acids), reducing the token rate while the decoder expands span-tokens to per-residue poses, and/or (ii) increasing the codebook size (e.g., 8k–64k) to lower per-token quantization error so that fewer tokens are needed at a target distortion. These options trade bitrate against utilization and latency, and remain compatible with our VQ training approach.

**Downstream 3D learning representation.** De-quantized code embeddings yield continuous, structure-aware features that potentially are more separable and clusterable than base GCPNet outputs, providing a strong encoder for downstream tasks, as observed in Yuan et al. (2025b).

**Generative modeling.** Because our structure representation is discrete and pose-invariant, it slots directly into autoregressive protein language model (PLM) pre-training: tokens can be interleaved with amino-acid symbols, letting us exploit next-token scaling laws observed in autoregressive Large

Language Models (LLM) (Kaplan et al., 2020). This unifies sequence and structure in a single model, enabling controllable generation via simple conditioning; e.g., sequence-given-structure and structure-given-sequence modes. Structure tokens serve as an explicit geometric prior for PLMs (Hayes et al., 2025; Su et al., 2023), potentially enabling more control over protein design. Moreover, the same machinery supports higher-throughput structure prediction from sequence by first predicting structural tokens and then mapping them to backbones with GCP-VQVAE's decoder (Pourmirzaei et al., 2025; Lu et al., 2024; Chen et al., 2024).

Several avenues look promising to explore: scaling GCP-VQVAE tokenization to multi-chain complexes, providing a lightweight variant for low-latency use, enriching tokens with side-chain information, and evaluating GenAI-friendliness of this new language to unify sequence and structure generation via autoregressive PLMs.

## 7 CONCLUSION

In this work, we have introduced GCP-VQVAE, a state-of-the-art open-source protein structure tokenizer. GCP-VQVAE offers improved structure reconstruction quality and generalization compared to prior methods, which could enable new (sequence and structure-based) predictive and generative modeling applications in future work.

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

## A  APPENDIX

### A.1  METHODS

Algorithms 1–3 define the decoder head, geometric losses, and alignment used in training. Given per-residue embeddings, the 6D head projects to two direction vectors and a translation, constructs a proper rotation via Gram–Schmidt, composes this rigid update with the running pose, and applies it to a fixed (N–$C_\alpha$–C) template to produce backbone coordinates. Reconstruction is supervised by the weighted sum in Equation 8: (i) aligned MSE after optimal Kabsch alignment, (ii) a clamped backbone distance-matrix loss, and (iii) a clamped pairwise orientation (direction) loss built from six backbone vectors per residue.

---

**Algorithm 1** Pseudocode for 6D rotation-based structure prediction

---

```
def dim6rot_structure_head(h: Tensor[N, d],
                           g_initial: Tuple[Tensor[N,3,3], Tensor[N,3,1]],
                           alpha: float = 1.0):
    """
    Predicts backbone coordinates from embeddings using a 6D rotation representation.
    This process is vectorized over N residues.

    h: [N, d] float32        -- input embeddings for N residues
    g_initial: (R_0, t_0)    -- initial SE(3) pose (running transformation)
    alpha: float             -- translation scaling hyper-parameter
    """
    # Define a fixed local reference frame for backbone atoms (N, C_alpha, C)
    X_local = torch.tensor(...) # Shape: [3, 3]

    # 1) Project embeddings to unconstrained translations and direction vectors
    # This represents the internal FFN and projection layers.
    h_ffn = LayerNorm(GELU(Linear(h)))
    # W_rigid projects h_ffn to 9 dimensions for t_tilde, a, and b
    rigid_proj = Linear(h_ffn) # Output shape: [N, 9]
    t_tilde, a, b = torch.split(rigid_proj, 3, dim=-1)

    # 2) Create rotation R_i via Gram-Schmidt orthonormalization
    a_hat = a / torch.norm(a, dim=-1, keepdim=True)
    c = torch.cross(a_hat, b, dim=-1)
    c_hat = c / torch.norm(c, dim=-1, keepdim=True)
    b_hat = torch.cross(c_hat, a_hat, dim=-1)

    # Stack column vectors to form the rotation matrix R
    R = torch.stack([a_hat, b_hat, c_hat], dim=-1) # Shape: [N, 3, 3]

    # 3) Scale the translation vector
    t = t_tilde * alpha # Shape: [N, 3]

    # 4) Form the local rigid body update g = (R, t) and compose it
    #    with the running pose g_initial = (R_0, t_0)
    R_0, t_0 = g_initial
    R_new = R_0 @ R
    t_new = (R_0 @ t.unsqueeze(-1)) + t_0

    # 5) Apply the new pose g_new to the local template to get final coordinates
    # Unsqueeze t_new for broadcasting over the 3 atoms in X_local
    X_pred = (R_new @ X_local.T).transpose(-1, -2) + t_new.transpose(-1, -2)

    return X_pred, (R_new, t_new) # Return coords and the updated pose
```

---

**Algorithm 2** Pseudocode for backbone *distance*, *direction* and *MSE* losses

```python
def compute_backbone_vectors(coords: Tensor[L, 3, 3]) -> Tensor[L, 6, 3]:
    """
    coords[..., 0/1/2] = (N, CA, C) atom positions for each residue.
    Returns six normalised vectors per residue:
        1)  N → CA             4)  -n_CA   (binormal of NCA plane)
        2)  CA → C             5)  n_N     (binormal of CN_prev-N plane)
        3)  C  → N_{i+1}       6)  n_C     (binormal of CA C N_{i+1} plane)
    """
    # bond vectors ----------------------------------------------------------
    v1 = coords[:, 1] - coords[:, 0]          # N → CA
    v2 = coords[:, 2] - coords[:, 1]          # CA → C
    v3 = pad_end(coords[1:, 0] - coords[:-1, 2])    # C → N_{i+1}
    v4 = -torch.cross(v1, v2)                 # -n_CA
    v5 = torch.cross(v3, v1)                  # n_N
    v6 = torch.cross(v2, v3)                  # n_C

    V = torch.stack([v1, v2, v3, v4, v5, v6], dim=1)
    return F.normalize(V, dim=-1)             # shape = [L, 6, 3]

def backbone_distance_loss(x_pred, x_true, mask):
    """
    x_pred/x_true : [L, 3, 3]  predicted & native backbone coords
    mask          : [L]        boolean, valid residues
    """
    P = x_pred[mask, :3].reshape(-1, 9)       # flatten 3 atoms → 9-D point
    T = x_true[mask, :3].reshape(-1, 9)

    D_pred = pairwise_l2(P, P)                # (N, N) distance matrix
    D_true = pairwise_l2(T, T)

    diff   = (D_pred - D_true).pow(2).clamp(max=25.)
    return diff.mean()

def backbone_direction_loss(x_pred, x_true, mask):
    """
    Uses six orientation vectors per residue as returned by
    `compute_backbone_vectors`.
    """
    V_pred = compute_backbone_vectors(x_pred[mask, :3])
    V_true = compute_backbone_vectors(x_true[mask, :3])

    # Pairwise dot-product tensors:  S_{ij}^{kl} = v_i^{(k)} · v_j^{(l)}
    S_pred = torch.einsum('ikd,jld->ijkl', V_pred, V_pred)
    S_true = torch.einsum('ikd,jld->ijkl', V_true, V_true)

    diff   = (S_pred - S_true).pow(2).clamp(max=20.)
    return diff.mean()

def aligned_mse_loss(x_pred, x_true, mask):
    """
    Mean-squared error after optimal rigid alignment
    (Kabsch) between predicted and native backbone coords.

    Args
    ----
    x_pred : Tensor[L, 3, 3]   # predicted (N, CA, C)
    x_true : Tensor[L, 3, 3]   # ground truth
    mask   : BoolTensor[L]     # valid residues
    """
    # 1) select masked residues and flatten atoms ------------- #
    P = x_pred[mask].reshape(-1, 3)   # (3N, 3)
    T = x_true[mask].reshape(-1, 3)   # (3N, 3)

    # 2) best-fit rotation / translation via Kabsch ------------ #
    R, t = kabsch(P, T)               # R ∈ SO(3), t ∈ ℝ³
    T_aln = (T @ R.T) + t             # align native coords

    # 3) mean-squared error ---------------------------------- #
    mse = ((P - T_aln).pow(2)).mean()
    return mse
```

**Algorithm 3** Pseudocode for rigid alignment via Kabsch

```
def kabsch_align(A: Tensor[N, 3],
                 B: Tensor[N, 3],
                 allow_reflections : bool = False):
    """
    Optimal rigid alignment (rotation R, translation t) of
    point cloud A onto reference cloud B.

    A, B : [N, 3] float32  -- corresponding coordinates
    """
    # 1) centre the clouds -------------------------------- #
    centroid_A = A.mean(0);  centroid_B = B.mean(0)
    AA = A - centroid_A     # centred source
    BB = B - centroid_B     # centred target

    # 2) SVD of covariance ------------------------------- #
    H = AA.T @ BB                            # 3x3
    U, _, Vh = torch.linalg.svd(H)          # H = U S Vh

    # 3) ensure proper rotation (det = +1) --------------- #
    if not allow_reflections and torch.det(Vh.T @ U.T) < 0:
        Vh[-1, :] *= -1

    R = Vh.T @ U.T                          # rotation
    t = centroid_B - R @ centroid_A         # translation

    return (R @ A.T).T + t                  # aligned A
```

## A.2 DATASETS

*Test A (low-confidence)* evaluates robustness under structural uncertainty using a curated subset of AlphaFold2 (AF2) models for SwissProt proteins. We enumerated all AF2-predicted structures available for SwissProt entries in AFDB and retained only those with average pLDDT $\leq 70$ (AFDB's 0–100 scale), discarding higher-confidence models. The surviving records were *cross-referenced* against the PDB to annotate the presence of experimental structures for the same proteins. Multimer entries were decomposed into per-chain monomer samples, after which we removed chains shorter than 25 amino acids and retained only chains with fewer than 25 consecutive missing residues. A final deduplication step produced 33 112 single-chain samples for Test A.

*Test B (species-diverse, taxonomic shift)* assesses generalization under distributional shift in species rather than structure confidence. We assembled a broad species roster (expanded to 100 and finalized at 97 species) and targeted roughly 2 500 proteins per species from AFDB without pLDDT filtering to reflect natural variability, dropping species with fewer than 1 000 available structures. After aggregation, per-chain conversion, and deduplication, this suite comprised 19 353 single-chain samples.

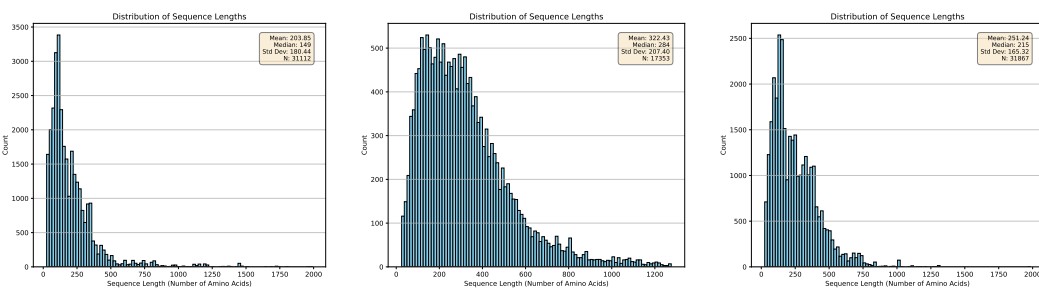

Figure 3: Distribution of sequence lengths in the three test sets: (left) Test set A, (middle) Test set B, and (right) Test set C. The majority of samples have sequence length $< 1280$ amino acids.

*Test C (high-confidence, under-represented taxa)* measures performance on high-confidence structures from taxa under-represented in SwissProt. We selected species with low SwissProt coverage and required high average pLDDT ($> 95$). The selected taxonomies were cross-referenced against

the PDB to find the presence of experimental structures for the proteins from those species. Applying the same preprocessing and deduplication as Test A above yielded 33 867 single-chain samples.

To obtain a balanced validation set independent of training, we first constructed three targeted hold-out suites and then set aside a fixed 2 000 randomly sampled chains from each to form a combined 6 000-chain validation set; the remaining examples constitute Test sets A, B, and C, respectively. The distribution of length sequences in the test sets is displayed in Figure 3.

From the zero-shot set, we defined interpretable structural subsets based on each protein's secondary-structure composition. Coil-rich (D) includes sequences with at least 60% coil/disordered content. Mainly $\alpha$ (A) requires an $\alpha$-helical fraction of at least 45% and an $\alpha - \beta$ difference of at least 15 percentage points; mainly $\beta$ (B) requires a $\beta$-strand fraction of at least 25% and a $\beta - \alpha$ difference of at least 10 points. Balanced (AB) captures mixed folds with $\alpha \geq 25\%$, $\beta \geq 15\%$, and $|\alpha - \beta| \leq 10$ points. We also report two data-quality subsets: sequences with no missing residues and sequences with at least one missing residue. Groups are defined independently (i.e., they may overlap).

## A.3 EXPERIMENTS

Table 5: Training hyperparameters for Stages 1 and 2.

| Hyperparameter | Stage 1 | Stage 2 |
|---|---|---|
| Optimizer type | AdamW | |
| $\beta_1$ | 0.9 | |
| $\beta_2$ | 0.98 | |
| $\epsilon$ | $10^{-7}$ | |
| Weight decay | $10^{-3}$ | |
| Gradient clipping (max $L_2$ norm) | 1.0 | |
| Gradient accumulation | 1 | |
| 8-bit parameters | ✗ | ✓ |
| Batch size (per GPU) | 16 | 4 |
| Learning-rate strategy | Cosine annealing | |
| Base learning-rate | $1e-4$ | $5e-5$ |
| Min learning-rate | $1e-6$ | |
| Warm-up steps | 16 000 | |
| Total steps | 375 000 | 1 500 000 |
| MSE coefficient ($\lambda_{\mathrm{MSE}}$) | $1 \times 10^{-3}$ | $5 \times 10^{-3}$ |
| Backbone distance coefficient ($\lambda_{\mathrm{dist}}$) | $10^{-2}$ | |
| Backbone direction coefficient ($\lambda_{\mathrm{dir}}$) | $5 \times 10^{-2}$ | |

This subsection compiles all experimental assets: Table 5 lists the full optimization schedule and training hyperparameters for Stages 1–2; Table 6 specifies the exact model configuration (GCPNet, VQ, and Transformer stacks). Figure 4 reports RMSD error distributions on CASP14/15/16 and CAMEO2024 across methods; Figure 5 shows qualitative robustness to contiguous missing-residue segments from benchmark structures; Figure 6 plots sequence length versus reconstruction RMSD with a least-squares fit on the validation set; Figure 7 visualizes the highest-RMSD (worst-case) examples per benchmark suite. Metrics use backbone atoms ($C_\alpha$) after Kabsch alignment, and statistics are over the full, unfiltered test sets.

We maintain the codebook with the exponential–moving–average (EMA) variant of VQ-VAE: for each code $k$ we keep decayed counts $N_k$ and sums $M_k$ of the encoder vectors assigned at the current step and update the entry by $\mathbf{e}_k \leftarrow M_k/(N_k + \varepsilon)$, with decay $\gamma$ (Stage 1: 0.99, Stage 2: 0.995; Table 6). Code entries are *not* updated by gradient; only the encoder receives the commitment penalty $\mathcal{L}_{\mathrm{commit}} = \beta \|\mathbf{z} - \mathrm{sg}[\mathbf{e}_k]\|_2^2$. We detect "dead" codes via an EMA count threshold (2 by default) and reinitialize them from recent encoder features using the same $k$-means seeding as at start-up. For the encoder pathway, we still use the rotation-trick Jacobian (Fifty et al., 2024) to pass informative gradients through the nearest-neighbour assignment; this affects only the encoder and does not modify the EMA update rule.

We set the coefficients $\lambda$ (Equation 8) by monitoring the per-term gradient norms and equalizing their contribution to shared parameters. This gradient-norm balancing prevents any single term (e.g., direction or distance) from dominating updates and yields stable, faster convergence.

Throughout all experiments, sequences longer than the model's input window are truncated to the first $L$ residues (Stage 1: $L = 512$, Stage 2: $L = 1{,}280$), and all metrics are computed on the truncated region.

Table 6: Configurations of GCP-VQVAE architecture.

| Hyperparameter | Stage 1 | Stage 2 |
|---|---|---|
| GCPNet Encoder | | |
| Initialization | (Jamasb et al., 2024) | Stage 1 |
| Input dimension (node scalars $h$) | 6 | |
| Input dimension (node vectors $\chi$) | 3 | |
| Input dimension (edge scalars $e$) | 8 | |
| Input dimension (edge vectors $\xi$) | 1 | |
| Hidden dimension (node scalars $h$) | 128 | |
| Hidden dimension (node vectors $\chi$) | 16 | |
| Hidden dimension (edge scalars $e$) | 32 | |
| Hidden dimension (edge vectors $\xi$) | 4 | |
| Number of layers | 6 | |
| SE(3) equivariance | ✓ | |
| Vector Quantization | | |
| Initialization | Random | Stage 1 |
| Hidden dimension | 256 | |
| Vocab size | 4 096 | |
| EMA Decay | 0.99 | 0.995 |
| Threshold EMA dead code | 2 | |
| Commitment weight ($\lambda_{\text{commit}}$) | 0.5 | 0.25 |
| Orthogonal reg weight ($\lambda_{\text{orth}}$) | 10 | |
| Orthogonal reg max codes | 512 | |
| Orthogonal reg active codes only | ✓ | |
| Rotation trick | ✓ | |
| KMeans initialization | ✓ | |
| KMeans iteration | 10 | |
| Transformer Encoder | | |
| Initialization | Random | Stage 1 |
| Hidden dimension | 1024 | |
| FF Multiplier | 4× | |
| Blocks | 12 | |
| Query heads | 12 | |
| Key-value heads | 3 | |
| Positional encoding | RoPE | |
| Query-key norm | ✓ | |
| Pre-norm | ✓ | |
| Transformer Decoder | | |
| Initialization | Random | Stage 1 |
| Hidden dimension | 1024 | |
| FF Multiplier | 4× | |
| Blocks | 16 | |
| Query heads | 16 | |
| Key-value heads | 1 | |
| Positional encoding | RoPE | |
| Query-key norm | ✓ | |
| Pre-norm | ✓ | |

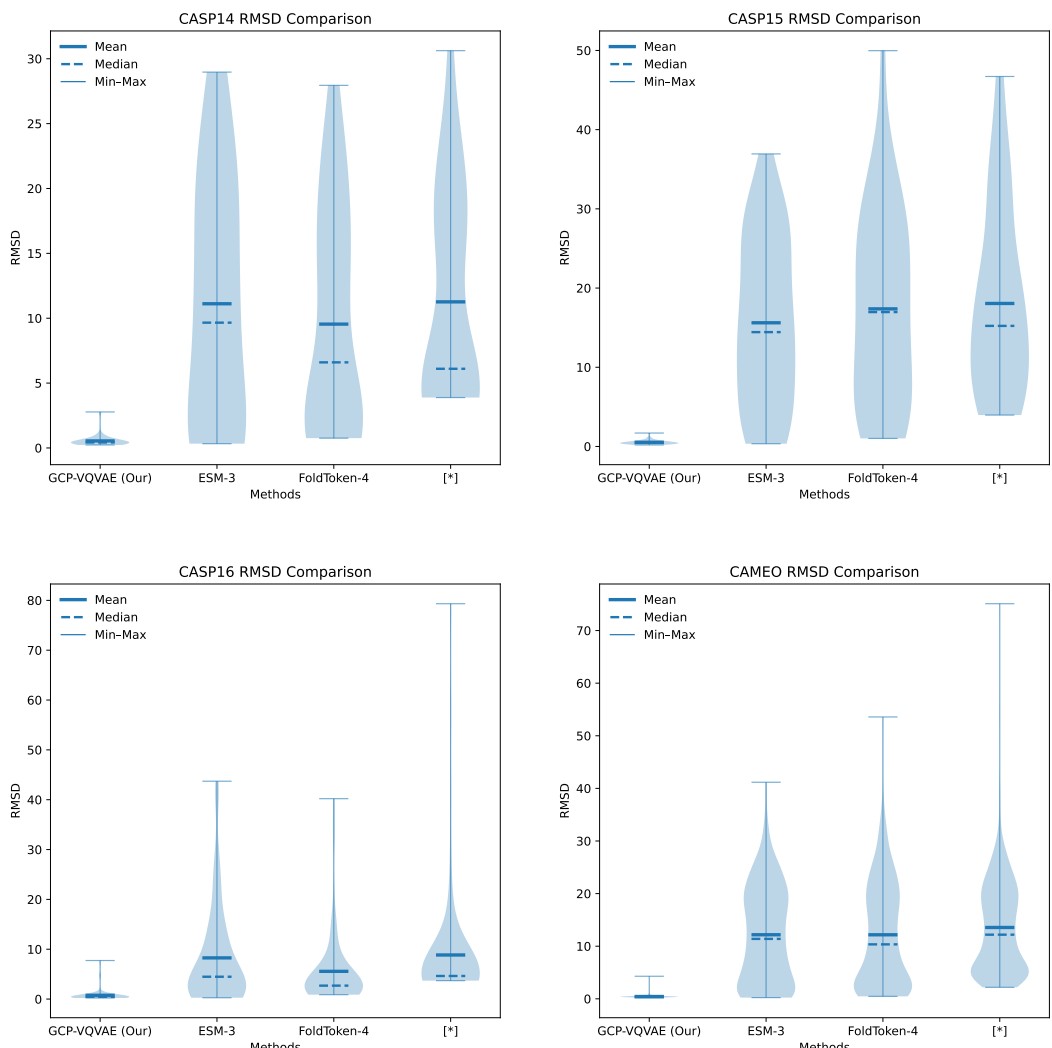

Figure 4: RMSD (Å) error distributions on CASP14, CASP15, CASP16, and CAMEO2024. Violin plots show the density of errors per method; horizontal markers denote the *mean* (solid), *median* (dashed), and *min–max* range (thin). Methods compared: GCP-VQVAE (ours), ESM-3 VQ-VAE, FoldToken-4, and the Structure Tokenizer of Gaujac et al. (2024) (shown as [∗]). For Gaujac et al. (2024), samples outside its supported length range (50–512 residues) are excluded; other methods use the full sets.

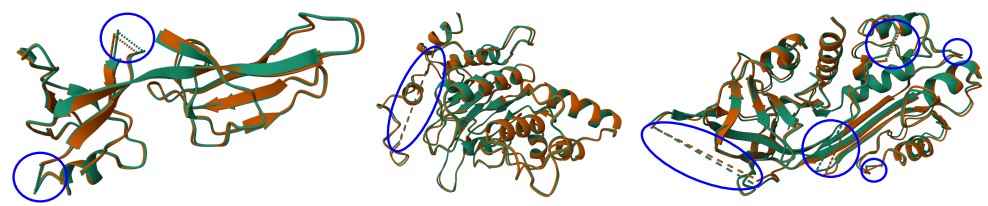

Figure 5: Superposition of GCP-VQVAE reconstructions (orange) with native backbones (green) for three proteins drawn from our external benchmark suites. Blue circles mark contiguous missing-residue segments (dashed guides span regions without deposited atoms); the model tracks the native structure outside the gaps.

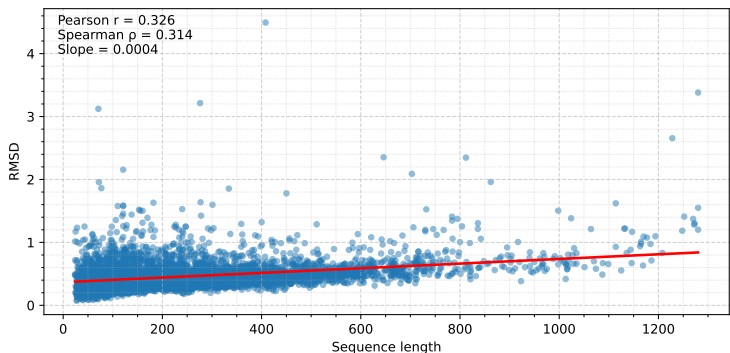

Figure 6: Sequence length vs. reconstruction error (validation set). Each point is one protein; the red line is a least-squares fit. The dependence on length is modest (Pearson $r = 0.326$, Spearman $\rho = 0.314$) with a small slope of $\approx 4 \times 10^{-4}$ Å/residue ($\sim 0.4$ Å per 1k residues).

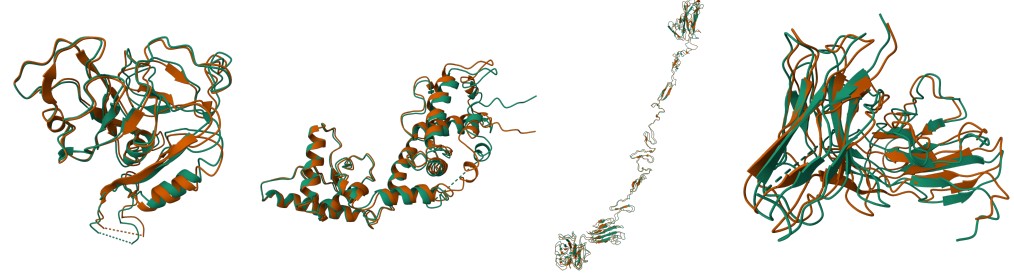

Figure 7: Superposition of GCP-VQVAE (orange) and native backbones (green) for the highest–backbone-RMSD example in each suite, left→right: CASP14, CASP15, CASP16, CAMEO2024.

## A.4 ZERO-SHOT PERFORMANCE

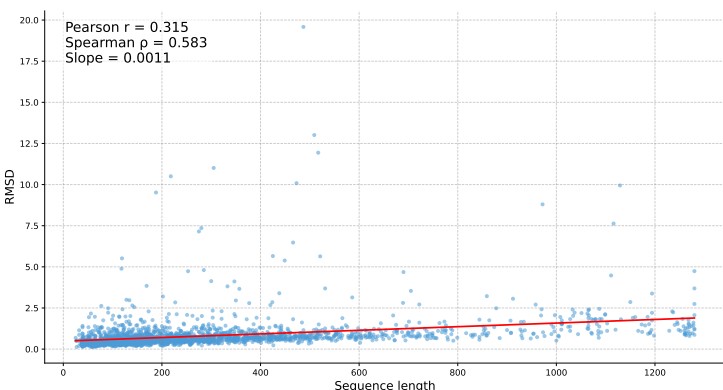

Figure 8: RMSD as a function of sequence length (number of amino acids) on the zero-shot set. Each dot is one protein. The red line is an ordinary-least-squares (OLS) fit; the panel reports Pearson $r$, Spearman $\rho$, and the fitted slope (RMSD units per residue).

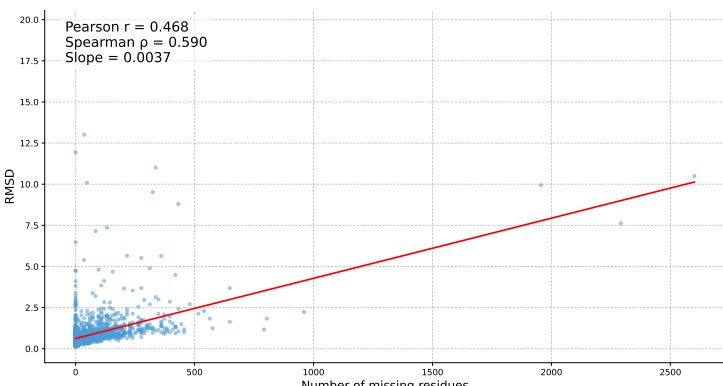

Figure 9: RMSD as a function of the number of missing residues on the zero-shot set. Points show individual proteins; the red line is an OLS fit. The panel reports Pearson $r$, Spearman $\rho$, and the fitted slope (RMSD units per missing residue).

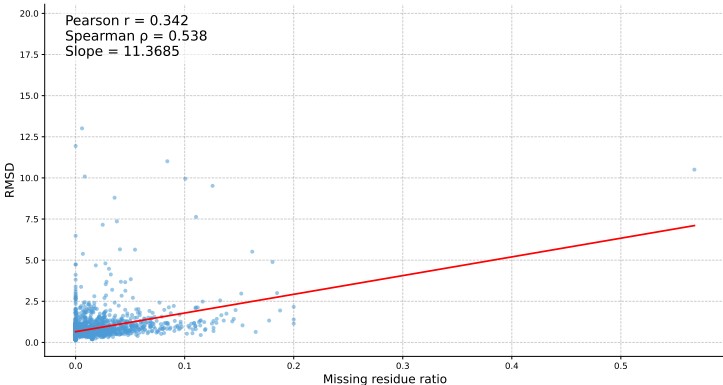

Figure 10: RMSD as a function of the missing-residue ratio on the zero-shot set. The red line is an OLS fit; the panel reports Pearson $r$, Spearman $\rho$, and the fitted slope (RMSD units per unit fraction; 1.0=100%).

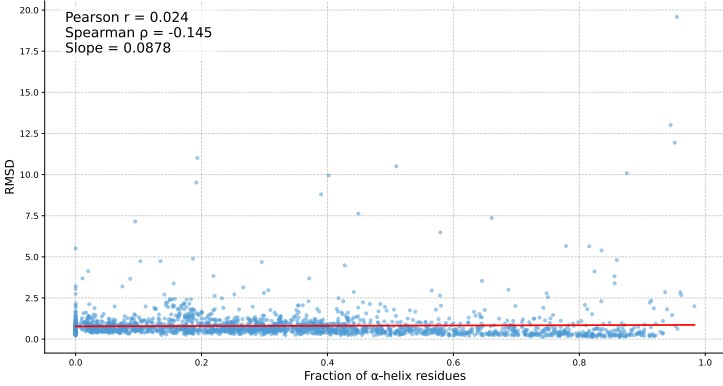

Figure 11: RMSD as a function of the fraction of $\alpha$-helix residues on the zero-shot set. The red line indicates an OLS fit. The panel reports Pearson $r$, Spearman $\rho$, and the fitted slope (RMSD units per unit fraction; 1.0=100%).

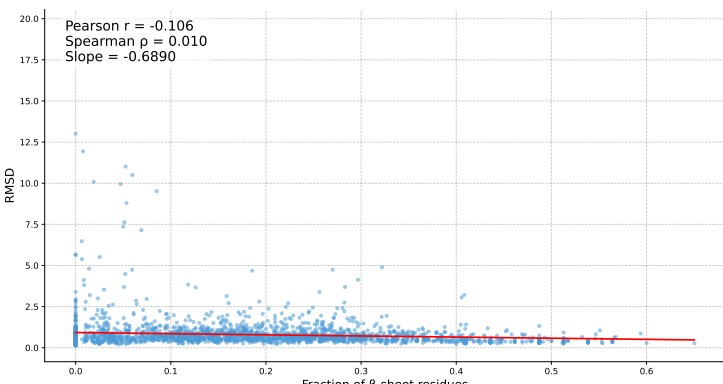

Figure 12: RMSD as a function of the fraction of $\beta$-sheet residues on the zero-shot set. The red line indicates an OLS fit. The panel reports Pearson $r$, Spearman $\rho$, and the fitted slope (RMSD units per unit fraction; $1.0=100\%$).

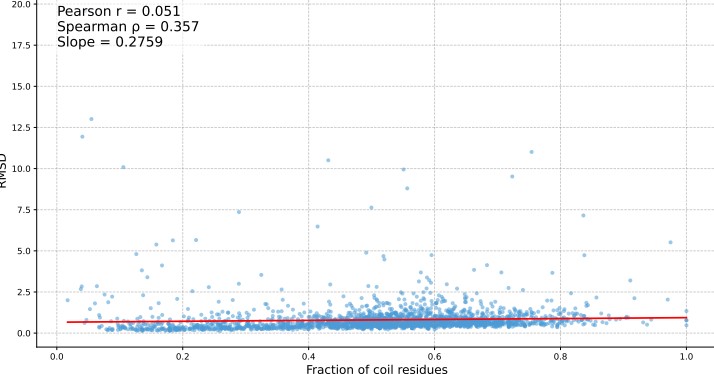

Figure 13: RMSD as a function of the fraction of coil residues on the zero-shot set. The red line indicates an OLS fit. The panel reports Pearson $r$, Spearman $\rho$, and the fitted slope (RMSD units per unit fraction; $1.0=100\%$).

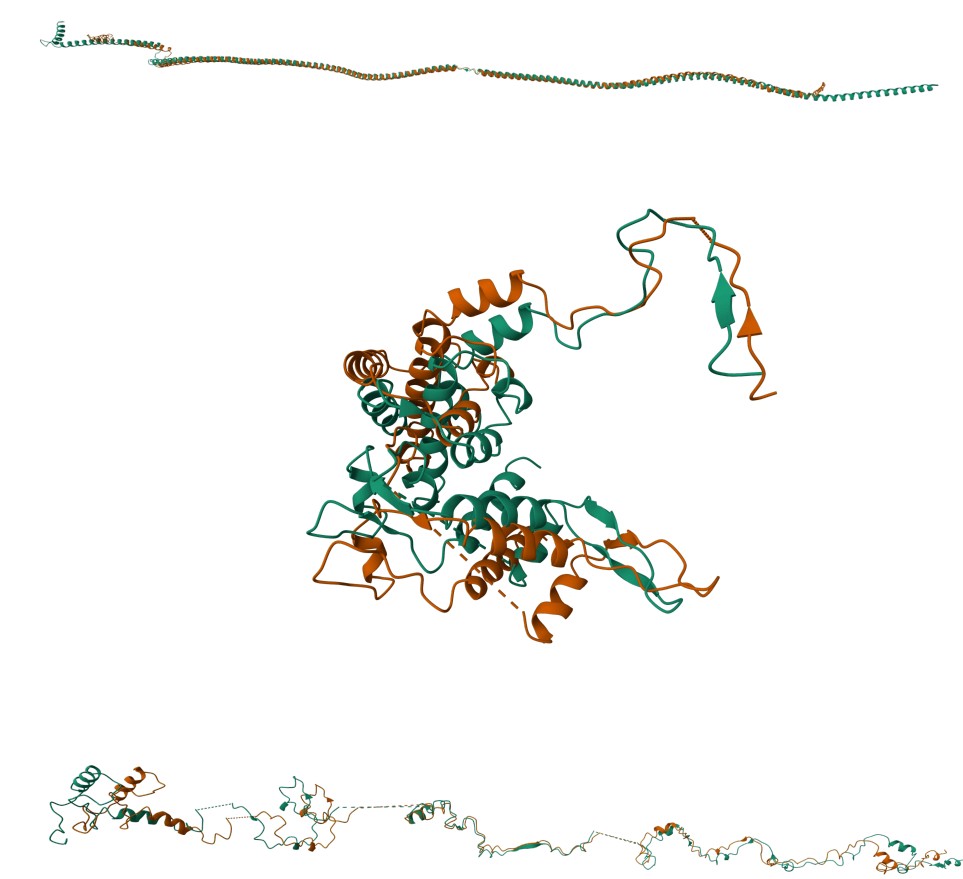

Figure 14: Worst zero-shot reconstructions (orange: GCP-VQVAE; green: native). Top: RMSD 19.5731 , TM-score 0.6935, length 487, NaN residues 0. Middle: RMSD 12.276 , TM-score 0.575, length 218, NaN residues 2601. Bottom: RMSD 11.0486 , TM-score 0.4766, length 305, NaN residues 336. The middle and lower cases exhibit an extreme number of NaN residues introduced by our current NaN-handling/augmentation pipeline, which forces the model to contend with long masked segments and degrades reconstruction accuracy; improving the NaN strategy (e.g., capping gap ratio, better segmentation, and length-aware masking) should mitigate this failure mode.

## A.5 COMPRESSION CALCULATION

Each residue is encoded by one code from a 4 096-entry book, i.e., $\log_2(4096) = 12$ bits = 1.5 bytes/residue. For $L = 512$ residues, the token stream is $512 \times 1.5 = 768$ bytes. Raw backbone coordinates (N, $C_\alpha$, C) stored as 32-bit floats require 3 atoms×3 coords×4 bytes = 36 bytes/residue, i.e., $36 \times 512 = 18{,}432$ bytes. If we include a tiny global pose header (e.g., rotation+translation) of $\approx 36$ bytes, the coded footprint is $768+36 = 804$ bytes. Thus the compression ratio is $18\,432/804 \approx 22.9\times$ (approximately $24\times$ if the pose header is omitted). This estimate concerns backbone only; metadata and format containerization add negligible overhead relative to the raw-float baseline.

## A.6 LLM USAGE

In this manuscript, we used large language models only for copy-editing: improving grammar, clarity, and style of author-written text.

