# OpenReview forum: "GCP-VQVAE: A Geometry-Complete Language for Protein 3D Structure"
_ICLR.cc/2026/Conference — Submitted to ICLR 2026_

### Official Review · Reviewer_t2D4 · 2025-10-25

**Soundness:** 3
**Presentation:** 3
**Contribution:** 2
**Rating:** 2
**Confidence:** 3

**Summary:**

They propose a VQVAE with a GCPNet encoder to tokenize protein structures. They train it on 24 million monomer backbone structures from the AlphaFold Protein Structure Database and evaluate reconstruction on several test sets, achieving state-of-the-art results.

**Strengths:**

- They achieve pretty good results in reconstruction when compared to FoldToken, ESM-3, and Gaujac et al 2024.
- Their experiments are fairly well-documented.

**Weaknesses:**

- They did not evaluate their work on generative tasks, so the actual usefulness for downstream tasks is questionable.
- There have been several recent works in the same area, including DPLM2, AiDO, and Bio2Token. You should do a comparison with these as well, or at least cite them.
- This line of work is generally very saturated. This paper does not sufficiently differentiate itself from other learned protein tokenizers.

**Questions:**

- Can you explain intuitively why you need the second and third reconstruction loss terms when you already have the MSE loss? Maybe an ablation here would be helpful too.
- Why did you use the original VQVAE formulation and not newer techniques like finite scalar quantization (FSQ), which significantly simplify  the vector quantization procedure?
- Any comment on the efficiency of your method? GCPNet uses explicitly quadratic pair representations, and so may be very memory intensive.

---

### Official Review · Reviewer_UGkJ · 2025-10-31

**Soundness:** 3
**Presentation:** 1
**Contribution:** 1
**Rating:** 2
**Confidence:** 5

**Summary:**

Authors presented a model named GCP-VQVAE, a new way to build a codebook for protein structure. This task could be used in building a sequence of protein structure, which could be used in  high-efficient structure search and pretraining.

**Strengths:**

The performance of GCP-VQVAE is higher than other current structure tokenization methods.

**Weaknesses:**

The novelty for using only another encoding model for protein structure is limited. The authors do not disscussion the potential use for this method.

**Questions:**

1. What are the explainability of the tokens, are similar token targeting similar structure, and different token have different structures?
2. What are the specific definition for the local environment of each residue
3. Can you add more ablation study of the GCP part?

---

### Official Review · Reviewer_bAXG · 2025-11-01

**Soundness:** 1
**Presentation:** 2
**Contribution:** 2
**Rating:** 2
**Confidence:** 4

**Summary:**

This paper presents GCP-VQVAE, a geometry-complete tokenizer for protein structures that uses an SE(3)-equivariant GCPNet encoder to preserve the directionality and chirality of protein backbones. By training a VQ-VAE on large-scale structural data, the authors discretize 3D protein geometry into a vocabulary of 4,096 structural tokens, achieving state-of-the-art reconstruction accuracy.

The key contribution lies in establishing a high-quality structural tokenization scheme, which forms a fundamental building block for protein representation learning. Such a well-trained structural vocabulary can facilitate future developments in protein structure–sequence modeling and generative protein design.

**Strengths:**

1. The paper addresses a highly valuable and practically important task — improving structure-aware tokenization for protein representation learning. The proposed model achieves strong results and demonstrates its potential to advance subsequent research on structural sequence modeling.
2. ESM-3 is generally recognized as a strong and widely used baseline for structure tokenization. The fact that the proposed method significantly outperforms ESM-3 indicates the model’s robustness and practical advantage in representing structural information.

**Weaknesses:**

1. Lack of comprehensive evaluation on structural tokenization quality.

The current evaluation which only reports RMSD and TM-score is not sufficiently comprehensive to support the claim that the proposed tokenization is optimal or generally effective. Structural tokenization can be evaluated from multiple dimensions — as highlighted in work such as Protein Structure Tokenization: Benchmarking and New Recipe (Yuan et al., 2025. "Protein structure tokenization: Benchmarking and new recipe.") — including:
- Downstream Effectiveness: Whether the learned tokens capture meaningful structure representations for supervised downstream tasks.
- Sensitivity: The model’s ability to distinguish between highly similar protein structures.
- Distinctiveness: The diversity and non-redundancy of codebook vectors.
- Codebook Utilization Efficiency: How effectively and evenly the token space is used, avoiding “dead tokens.”
It is recommended that the authors evaluate the model under a more comprehensive benchmark that covers these four aspects, ideally under comparable experimental settings.

2. Insufficient ablation or interpretability analysis.

The model seems to use a relatively standard GCP-style architecture, but achieves surprisingly large improvements over ESM-3. Without ablation studies or detailed analysis, it remains unclear what design choices or factors lead to this improvement. An investigation into architectural variants, codebook configurations, or structure-guided loss functions would greatly strengthen the paper.

**Questions:**

The DPLM2 model (Hsieh et al., 2025. "Elucidating the design space of multimodal protein language models.") is another recent and representative baseline in multimodal protein language modeling. How does the proposed method compare to DPLM2 in similar structural tokenization task?

---

### Official Review · Reviewer_a8Ba · 2025-11-01

**Soundness:** 3
**Presentation:** 3
**Contribution:** 2
**Rating:** 2
**Confidence:** 4

**Summary:**

GCP-VQVAE is an open-source protein tokenizer that improves structure reconstruction quality over prior methods. It achieves backbone RMSD of 0.8033 Å and a TM-score of 0.9747 on over 2000 completely new experimental structures

**Strengths:**

- Low reconstruction error, allows for efficiently compressing structures (practical lossy backbone codec)
- Open source, reproducible, efficient
- Well written, the steps taken for development are clearly explained

**Weaknesses:**

- The technical contribution of the wok is unclear: both GCP and VQVAE were developed before
- The evaluation is performed only on the reconstruction and in the "perfect" setting: how robust would be the results upon introducing some noise or randomness in the input structure?

**Questions:**

- The paper "Protein structure tokenization: Benchmarking and new recipe" from ICML is cited but not discussed; what are the limitations? why did the authors chose not to use the proposed benchmark?
- It would be useful to understand the relationship between the structural encoding and the backbone phi/psi angle clustering in the Ramachandran plot. GCP-VQVAE seems like a method to group populous regions in Ramachandran plots into 4096 groups. Is it the case in practice?
- Beyond 3D structure reconstruction, do the inferred structural tokens provide some advantage over previous methods for structural alignment, remote homology search, mutational outcome prediction, or any other relevant downstream task?
- Could the authors clarify the methodological advance brought by this work?

---

### Meta-Review · Area_Chair_749S · 2026-01-03

**Summary:**

The submission proposes a tokenizer for protein 3D structures, with a particular consideration for reflecting chirality by using the "geometry-complete" perceptron architecture. The high-level framework for learning the tokenizer is based on VQ-VAE.

Reviewers expressed appreciation on the significance of the target task and the impressive reconstruction performance. But they all mentioned the insufficiency of limited evaluation aspects, i.e., the authors only showed reconstruction performance (Reviewer bAXG provided a detailed guidance for more dimensions; especially for representation utilities), and the concern about the technical novelty and comparison with existing highly relevant tokenizers that are of community's interest.

**Reviewer Concerns:**

The authors did not provide a rebuttal. I regard the concerns raised by the reviewers indeed accounted as the insufficiencies of quality and contribution of this submission.

**Reviewer Scores:**

The authors did not provide a rebuttal.

---

### Decision · Program_Chairs · 2026-01-26

Reject